# THINC-Integrated Tool (THINC-it): A Brief Measurement of Changes in Cognitive Functioning and Its Correlation with the Life Quality of Patients with Schizophrenia and Related Disorders—A Pilot Study

**DOI:** 10.3390/brainsci13030389

**Published:** 2023-02-24

**Authors:** Joanna K. Szmyd, Karol Lewczuk, Kayla M. Teopiz, Roger S. McIntyre, Adam Wichniak

**Affiliations:** 1Third Department of Psychiatry, Institute of Psychiatry and Neurology, Jana III Sobieskiego 9, 02-957 Warsaw, Poland; 2Institute of Psychology, Cardinal Stefan Wyszynski University, Dewajtis 5, 01-815 Warsaw, Poland; 3Brain and Cognition Discovery Foundation, 73 Mathersfield Drive, Toronto, ON M4W 3W4, Canada; 4Department of Psychiatry, University of Toronto, 27 King’s College Cir, Toronto, ON M5S 1A1, Canada; 5Department of Pharmacology, University of Toronto, 27 King’s College Cir, Toronto, ON M5S 1A1, Canada

**Keywords:** schizophrenia, cognitive functions, quality of life, cognition, RDoC, domains, depression, patient-reported outcomes, function, measurement-based care

## Abstract

Background: This pilot study aimed to assess patients’ cognitive functioning with the Polish version of the THINC-it tool and to analyze its association with self-reported quality of life (QOL). Methods: Twenty-one patients (mean age: 37.8 ± 10.4) were assessed at baseline and after six weeks of a standard therapeutic outpatient program. Participants completed the World Health Organization QOL Questionnaire (WHOQOL-BREF) and the THINC-it tool at both visits. The tool consists of tasks evaluating working memory (SYMBOL CHECK), attention (SPOTTER), executive functions (TRIALS), and cognitive skills (CODEBREAKER). Results: During the second visit, patients showed significant improvements in mean latency of correct responses of SPOTTER: *p* = 0.021, Cohen’s d = 0.38 and in the Physical health domain: *p* = 0.007, Cohen’s d = 0.37. The number of correct responses for CODEBREAKER was positively associated with the Physical health domain at visit 1 (r = 0.53, *p* = 0.014) and visit 2 (r = 0.42, *p* = 0.058). The number of correct responses at SYMBOL CHECK was positively related to QOL in the Environment domain only at visit 2 (r = 0.45, *p* = 0.042). Conclusions: These results suggest the THINC-it tool has utility as a cognitive measure in adults with schizophrenia in both clinical and research settings.

## 1. Introduction

A compelling body of research clearly shows the extent of cognitive impairments experienced by individuals with schizophrenia and related disorders. Patients experience deficits in memory, attention span, language and motor skills, and executive functions [1]. The persistent nature of cognitive dysfunctions is frequently seen as the most chronically disabling component of schizophrenia [2]. The chronic persistence of cognitive deficits and their negative impact on the functioning of patients with schizophrenia provide the impetus to characterize cognitive deficits more fully and their impact on patient-reported outcomes (PROs). Convergent data indicate that the difficulties associated with dysfunctions in this area may also persevere during the period of remission of schizophrenia [2]. The persistence of cognitive deficits during symptomatic remission in patients mediates functional impairment (e.g., studying, working, or maintaining social activity) and significantly reduces their level of satisfaction with life [3].

The World Health Organization (WHO) defines quality of life as the individual’s perception of their position in life, in the context of the culture and value systems in which they live, and in relation to their goals, expectations, standards, and concerns [4]. Cognitive dysfunctions have great importance for social and occupational performance and oftentimes might have a more significant impact on quality of life than the severity of psychotic symptoms [5,6,7]. Several previous studies show that executive functioning is especially a strong predictor of the general quality of life. More severe cognitive deficits are associated with poor quality of life in patients with schizophrenia spectrum disorder [8,9,10]. The foregoing observation (e.g., the direct mediational effect of cognition on PROs) has also been observed in mood disorders [11,12].

It has been proposed that the disruption in fundamental cognitive processes is the core source of many symptoms that occur in schizophrenia spectrum disorder. Goldman-Rakic and Selemon proposed that deficits in working memory and the inability to process information from the external environment are fundamental disturbances in schizophrenia [13], and a similar hypothesis was proposed by Hemsley [14]. Cognitive dysfunctions, the inability to coordinate the perception, encoding, retrieval, and prioritization of information have their roots in structural and functional brain abnormalities. Morphometric studies in persons living with psychosis have reported reductions in grey matter volume in select brain regions [15,16]. More specifically, the previous study has shown the link between the volume and thickness of the prefrontal cortex and executive function disruption. Working memory deficits were also linked with smaller hippocampal grey matter volume and episodic memory, which is highly dependent on the hippocampus in healthy individuals, and might be impaired in schizophrenia patients due to the reduction in hippocampal volume [17,18].

Despite several attempts to develop novel drugs for the treatment of cognitive impairment in schizophrenia in the last years [19], still, no pharmacological treatment is registered for this indication. Emerging evidence indicates that cognitive remediation may be capable of improving measures of cognitive functions in persons living with schizophrenia [20,21]. Improvements in cognitive functions have been associated with increased activation of the inferior frontal gyrus, and increased activation in the frontal, parietal, inferior frontal junction, and visual cortex [20]. These findings were replicated by separate groups who also observed increases in left lateral prefrontal activation [21,22]. Improvement of the medial prefrontal activation pattern was also observed after cognitive remediation with word generalization tasks and recognition tasks. Mechanistic research attempts to identify substrates that subserve cognitive functions and mechanisms of therapeutic action in patients with psychotic disorders [23,24]. Notwithstanding the clinical relevance of cognitive deficits of persons living with schizophrenia, there is a pressing need for a brief, no-cost, digitalized, point-of-care cognitive assessment tool that integrates both subjective and objective measures of cognition. Dementia screening tools, e.g., Mini-Mental Status Examination [25] are not appropriate, as they are insensitive to subtle deficits and are constrained by ceiling effects. Moreover, there is a need for a cognitive tool that not only can capture the multi-dimensional cognitive deficits but can also be sensitive to change across time with or without treatment.

The THINC-it tool is an easy-to-use mobile application (THINC-it) that can be used to screen and measure cognitive functions not only in patients suffering from depression [26,27]. This tool was developed by the THINC-it Task Force (http://thinc.progress.im accessed on 21 February 2023)—it is digital and is composed of well-known cognitive paradigms that are focused on key cognitive areas such as working memory, attention span, and executive functions. More specifically, this tool consists of four tasks: “SYMBOL CHECK” is based on a one-back paradigm and is a measure of working memory; “SPOTTER” is a choice reaction task that is a measure of attention; “TRIALS” is a variation of the Trail Making Test (TMT), part B, for executive functions; and “CODEBREAKER”, which was created on the basis of the Digital Symbol Substitution Test (DSST) paradigm that also occurs as the Digit Symbol-Coding test in the Wechsler Adult Intelligence Scale (WAIS-R). Performance in these tasks is dependent on the functional integrity of working memory, attention, executive functions, and other cognitive skills [28]. This set is completed by a subjective measure of cognitive function—Perceived Deficits Questionnaire, 5 items (PDQ-5). The questionnaire includes items addressing such areas as performance regarding the concentration of attention span, planning and organization, and retrospective and prospective memory [29,30].

Validation studies from the THINC-it Task Force revealed that the THINC-it tool is capable of detecting cognitive deficits in persons with major depressive disorder as well as sensitivity to change with treatment [26,27]. Furthermore, a separate analysis and healthy controls were conducted to examine the psychometric characteristics of THINC-it. To evaluate temporal reliability, stability, and convergent validity, a group of 100 participants completed the full set of cognitive assessments. For each THINC-it test (Spotter, Symbol Check, Codebreaker, Trials) an appropriate comparison test was administered accordingly (Identification Task, OBK—One-Back Memory task, DSST—Digit Symbol Substitution Test, TMT-B—Trail Making Test Part B). This study showed that the THINC-it tool has high levels of reliability (temporal reliability—Pearson’s r correlations varying between 0.75 and 0.81; Intrarater reliability—for all tests ranged between 0.7 and 0.93) and stability (standard deviation values for accuracy measures ranging from 5.9 to 11.23, and for latency measures ranging from 0.735 to 17.3). In addition, levels of convergent validity were in the acceptable range. The correlations between the Spotter and Identification Task of 0.44; Trials and TMT-B of 0.74; and the Codebreaker and DSST of 0.63 indicate that those THINC-it tasks are a justifiable proxy measure of the tasks from the comparison assessment. This trend has not occurred between the Symbol Check task and the OBK—One-Back Memory task, which resulted in a considerably lower correlation (r = 0.19) [26].

Separate analysis with the THINC-it tool revealed that deficits in cognitive functions were highly associated with deficits in self-rated quality of life and functioning in persons with major depressive disorder [31].

Herein, we sought to determine whether the Polish version of the THINC-it tool was capable of detecting cognitive deficits in individuals living with schizophrenia and whether deficits in cognitive functions correlated with self-reported quality of life, as measured by The World Health Organization Quality of Life Questionnaire (WHOQOL-BREF). We were additionally interested in determining whether the THINC-it tool could be administered as a repeat measure in the schizophrenia population with sensitivity to change across time.

## 2. Materials and Methods

### 2.1. Participants

The study protocol was approved by the Bioethical Committee of the Institute of Psychiatry and Neurology in Warsaw and was carried out in accordance with the Declaration of Helsinki (approval number: 1/2021). The study included patients from the Daily Ward. Twenty-one participants were included in the study (7/14 females/males; mean age: 37.8 ± 10.4) among which 15 were diagnosed with schizophrenia, 3 were referred to the ward with a preliminary diagnosis of schizophrenia and were finally diagnosed with affective disorders with psychotic features (two patients with bipolar disorder and one with major depressive disorder), and 3 were diagnosed with schizophrenia-like disorders.

All subjects were carefully screened and examined for current mental conditions and severity of symptoms to exclude patients with acute psychotic symptoms, side effects of pharmacological treatment, or somatic disorders that could negatively influence the cognitive testing. All patients were introduced to the study procedure by a psychologist and wished to participate in the study to evaluate their cognitive functions and quality of life. Due to the exclusion criteria, the number of participants was restricted, which, alongside the heterogeneous nature of the group, added to the limitations of the project.

### 2.2. Procedure and Measures

The Polish version of the THINC-it tool was used for the evaluation of the level of cognitive functions. A team of psychologists adapted the THINC-it tool to the Polish language. The short version of the survey assessing quality of life—WHOQOL-BREF was used to measure patients’ quality of life. WHOQOL-BREF consists of 4 domains: Physical health, Psychological, Social Relationships, and the Environment.

The study consisted of 2 visits. During each visit, patients were asked to carry out the THINC-it tool twice (iPad version) and to fill in the WHOQOL-BREF Questionnaire only once (paper–pencil version). The first performance of THINC-it on each visit was considered as training, and the results of the second performance were used as an assessment of cognitive functions and, therefore, named Assessment 2 in Table 1. The second visit took place approximately after six weeks following the first visit and included the same procedure. Between both visits, patients took part in a therapeutic program at the day ward, approximately six hours per day, which included various types of non-pharmacological interventions (including basic cognitive functions training, metacognitive training, social skills training, and CBT therapy).

### 2.3. Analysis

Means, standard deviations, and ranges were reported for each assessment. The significance of changes observed between particular assessments was assessed by paired *t*-test and size effect. The relationship between the results of the four THINC-it tasks and four WHOQOL-BREF domains was tested with the Pearson’s r correlation coefficient test. *p*-values of <0.05 (two-sided for *t*-tests) were considered significant. Cohen’s d was used as a measure of the effect size [32].

## 3. Results

Table 1 depicts descriptive statistics (along with 95% Confidence Intervals) for two assessment sessions in each of the THINC-it tasks. For the mean latency for correct responses in the Spotter task, we have noted a significant improvement, with a medium effect size (Cohen’s d). Additionally, some improvement between the first and second visits was also observed for other THINC-it tasks; however, the corresponding comparisons did not reach significance, and the obtained effect sizes were modest (Table 1).

As compared with data published for healthy controls from a reference study (Table 1), the N-back type task (Symbol Check), the symbol substitution test (Codebreaker), and the Spotter task posed the greatest challenge for the patients [26].

Quality of life indices for the first and second visits are depicted in Table 2. As compared with the measurement at visit 1, patients obtained higher scores in quality of life in the Physical health domain at the second visit measurement (medium effect size; Table 2). The quality of life scores in the Psychological and Environment domains also improved; however, corresponding comparisons, based on the current, rather small pilot study sample (*n* = 21), led to non-significant findings.

The analysis of the relation between THINC-it tasks and WHOQOL-BREF domains of quality of life showed that the number of correct responses in the Codebreaker task was positively associated with the quality of life scores in the Physical health domain at visit 1 (r = 0.53, *p* = 0.014). The correlation obtained at visit 2 was non-significant at slightly over the significance level (r = 0.42, *p* = 0.058).

The number of correct responses in the Symbol Check task was positively related to the quality of life scores in the Environment domain but only at visit 2 (r = 0.45, *p* = 0.042).

As presented in the Table 3, the number of correct answers in the Codebreaker task at visit 1 is positively related to the quality of life scores in the Physical health domain at visit 1 (r = 0.053, *p* = 0.014) and visit 2 (r = 0.42; *p* = 0.058). The number of correct answers in the Symbol Check task in the second visit was positively associated with the quality of life scores in the Environment domain in visit 2 (r = 0.45; *p* = 0.042). For other measures, better cognitive functioning was regularly associated with higher quality of life, but the comparisons did not exceed the limit of statistical significance due to the small number of respondents (*n* = 21).

## 4. Discussion

This pilot analysis replicates other lines of research that cognitive impairment in schizophrenia is highly associated with quality of life (Physical health and Environment domains). We also observed that cognitive symptoms improved across visits. This observation is in keeping with other results where therapeutic interventions such as cognitive remediation [33], metacognitive training [34], or virtual reality training [35] provide an improvement in cognitive functions, which can be reasonably expected in persons with schizophrenia.

Furthermore, our study showed only modest improvement in cognitive functions and quality of life in patients with schizophrenia spectrum disorders after six weeks of standard care at point-of-care. None of the persons in our study, however, received experimental pro-cognition strategies and/or cognitive remediation.

Our ability to analyze the obtained data is limited, mainly due to the small number of subjects in the investigated group and the short observation period. This significantly reduced our ability to set conclusions about the relationships between cognitive functioning and quality of life. A short follow-up period also reduced our ability to form firm conclusions about the course of cognitive deficits in a chronic disorder. Despite this limitation, we did detect a significant relationship between the number of correct responses in Codebreaker tasks and an association with quality of life (i.e., Physical health domain) across both study visits.

We are also aware that the results observed in our research cannot be compared with those of specialized cognitive remediation programs.

Due to the limitations of the study, we were not able to gather more participants in the project that would stand as a healthy control group. Therefore, to add additional context, we have used data for healthy controls from the reference study of Harrison et al., 2018. We are aware that this is not fully appropriate to make conclusions, and that ideally, future projects should consist of healthy controls for proper comparison purposes.

Nonetheless, observational data like ours, in real-world patients, at a representative treatment program, provide the impetus to establish the effectiveness of cognitive remediation and its contribution to the general well-being of patients and their day-to-day functioning.

In addition to its clinical application, the THINC-it tool is used commonly across research studies to measure and monitor cognitive impairment in Major Depressive Disorder—MDD [36,37,38,39,40,41]. Studies show the associations between cognitive deficits and global and specific psychosocial deficits in patients with current and remitted MDD [42,43,44]. THINC-it was also used to explore connections of cognitive dysfunctions with psychosocial functioning and as a tool for detecting cognitive dysfunction amongst adults with MDD who also experience pain [31,43]. This highlights the clinical value and interpretability of the THINC-it tool as a cognitive screening device in patients with MDD.

To our best knowledge, our pilot study is also the first to document the use of THINC-it as a cognitive measure in patients with schizophrenia. We are convinced that an extended analysis of the convergent and divergent validity of the Polish version of the THINC-it tool is needed, and this is an aim for future studies—ideally based on a larger sample of participants with the ability to extend the period of therapeutic interventions and postponed assessment of patients’ cognitive performance. This tool is a free-of-charge, easy-to-use mobile application that is practical to implement and provides clinically relevant information.

It was observed by our group that most patients viewed the Codebreaker task as the most demanding; these complaints are understood as Codebreaker addresses executive functions, working memory, and speed of processing—three cognitive functions in which patients with schizophrenia usually show some deficits. We think that some small adaptation of the two more difficult tasks—Codebreaker and Symbol Check—could be beneficial to allow more cognitively disabled patients with schizophrenia to use it as well. Future studies should validate the THINC-it tool in persons with schizophrenia in a larger sample; validating its ability to detect change over time, with or without treatment, would be highly relevant as well.

Taken together, cognitive deficits in persons with schizophrenia are persistent, often progressive, and mediate functional impairment and quality of life deficits. Our preliminary data suggest that the THINC-it tool can be capable of detecting cognitive deficits in persons with psychotic disorders and may represent a potential clinical and research outcome measure. Larger sample sizes and real-world samples will be instructive; interoperability with electronic health records as well as other technology increasingly being employed in the assessment, monitoring, and treatment of persons with mental disorders will also be important to have.

## Figures and Tables

**Table 1 brainsci-13-00389-t001:** Descriptive statistics and 95% Confidence Interval based on Standard Error of the mean for THINC-it cognitive measures, along with corresponding paired *t*-Test comparisons. Data for patients with schizophrenia participating in the current study (*n* = 21), as well as for healthy controls from a separate study by Harrison et al., 2018 [26].

	Patients with Schizophrenia				Healthy Controls from Reference Study Harrison et al., 2018 [26]
THINC-it Task	Visit 1, Assessment 2 Mean ± SD; 95% CI; Range	Visit 2, Assessment 2 Mean ± SD; 95% CI; Range	*t*-Test t Value and *p*-Value	Effect Size (Cohen’s d)	Visit 2, Values for Study Harrison et al., 2018 Mean and 95% CI (Based on Standard Error of the Mean)
Spotter (mean latency for correct responses in msec)	806.90 ± 203.68 (95% CI: 762.46–851.35); Range: 760	725.71 ± 222.96; (95% CI: 677.06–774.37); Range: 842	t(20) = 2.51; *p* = 0.021	0.38	577 (95% CI: 545–609)
Symbol Check (number of correct responses)	17.10 ± 9.80; (95% CI: 14.96–19.23); Range: 34	19.48 ± 9.28; (95% CI: 17.45–21.50); Range: 27	t(20) = −1.84; *p* = 0.081	0.25	29 (95% CI: 27–31)
Codebreaker (number of correct responses)	41.57 ± 13.88; (95% CI: 38.54–44.60); Range: 53	45.24 ± 16.34; (95% CI: 41.67–48.80); Range = 69	t(20) = −1.28; *p* = 0.215	0.24	69 (95% CI: 65–73)
Trials (time taken for completion in sec)	34.95 ± 14.82; (95% CI: 31.71–38.18); Range: 68.27	32.08 ± 10.68; (95% CI: 29.74–34.41); Range = 37.45	t(20) = 0.97; *p* = 0.345	0.22	27 (95% CI: 23–32)

**Table 2 brainsci-13-00389-t002:** Descriptive statistics: means and standard deviations for the quality of life scores (WHOQOL-BREF domains) from visit 1 and visit 2.

	Visit 1	Visit 2	*t*-Test	Effect Size
WHOQOL Domain	Mean ± SD	Mean ± SD	t-Value and *p*-Value	Cohen’s d
Physical Health	49.19 ± 19.97	56.42 ± 18.65	t(20) = −3.03; *p* = 0.007	0.37
Psychological	57.23 ± 19.39	59.9 ± 17.05	t(20) = −1.22; *p* = 0.238	0.15
Social Relationships	61.28 ± 14.32	61.28 ± 20.62	t(20) = 0.00; *p* = 1.00	0
Environment	65.61 ± 12.38	68.23 ± 11.78	t(20) = −1.56; *p* = 0.136	0.22

**Table 3 brainsci-13-00389-t003:** Correlations for THINC-it cognitive measures and for the quality of life scores (WHOQOL-BREF domains) from visit 1 and visit 2 (Pearson’s r, *p*-values in brackets).

	WHOQOL Domain							
	Physical Health (Visit 1)	Psychological (Visit 1)	Social Relationships (Visit 1)	Environment (Visit 1)	Physical Health (Visit 2)	Psychological (Visit 2)	Social Relationships (Visit 2)	Environment (Visit 2)
Spotter (visit 1)	−0.26 (0.253)	−0.08 (0.719)	0.07 (0.76)	0.02 (0.932)	−0.35 (0.126)	−0.30 (0.185)	−0.33 (0.147)	−0.01 (0.983)
Symbol Check (visit 1)	−0.03 (0.903)	−0.07 (0.75)	−0.10 (0.664)	0.29 (0.21)	−0.00 (0.997)	0.15 (0.527)	0.26 (0.25)	**0.45 * (0.042)**
Codebreaker (visit 1)	**0.53 * (0.014)**	0.15 (0.522)	−0.22 (0.342)	0.07 (0.754)	0.42 (0.058)	0.19 (0.414)	−0.06 (0.789)	0.20 (0.375)
Trials (visit 1)	−0.23 (0.319)	0.03 (0.9)	0.10 (0.653)	−0.01 (0.953)	−0.21 (0.372)	−0.10 (0.656)	−0.24 (0.299)	−0.10 (0.66)
Spotter (visit 2)	−0.17 (0.454)	−0.03 (0.914)	−0.08 (0.718)	0.09 (0.693)	−0.13 (0.578)	−0.09 (0.699)	−0.28 (0.217)	0.27 (0.23)
Symbol Check (visit 2)	0.12 (0.618)	−0.02 (0.919)	−0.07 (0.777)	0.43 (0.054)	0.13 (0.571)	0.21 (0.359)	0.43 (0.053)	**0.44 * (0.047)**
Codebreaker (visit 2)	**0.45 * (0.04)**	0.11 (0.624)	−0.04 (0.863)	0.17 (0.467)	**0.44 * (0.047)**	0.22 (0.331)	0.09 (0.698)	0.22 (0.328)
Trials (visit 2)	−0.41 (0.063)	−0.15 (0.513)	0.08 (0.722)	0.07 (0.772)	−0.41 (0.066)	−0.25 (0.269)	−0.27 (0.232)	−0.00 (0.989)

Note. Significant relationships are bolded and marked with *.

## Data Availability

The data presented in this study are available on request from the corresponding author, email: jk.szmyd@gmail.com.

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
