# Peer review of "THINC-Integrated Tool (THINC-it): A Brief Measurement of Changes in Cognitive Functioning and Its Correlation with the Life Quality of Patients with Schizophrenia and Related Disorders—A Pilot Study"

_brainsci, 2023, doi:10.3390/brainsci13030389_

Round 1
Reviewer 1 Report
Review:
Title: THINC-Integrated Tool (THINC-it): A Brief Measurement Of Changes In Cognitive
Functioning And Its Correlation With The Life Quality Of Patients With Schizophrenia And
Related Disorders – A Pilot Study.
Authors: Szmyd, J. K. et al
Table 1 – healthy controls: I did not have a chance to check the description of data, but I am not
sure if two groups are similar to each other for all aspects, except schizophrenia. Otherwise, this
study cannot be an appropriate comparative study. What was the sample size of the control
group?
Another point is, the sample size is too small. With this sample size, authors might lose the
power. It would be better, if the sample size is increased, or if the reason for having a small
sample is justified. I also have a concern on verification of assumptions for hypothesis testing for
the correlations.
Correlations from the physical health domain: visit 1 had r = 0.53 with p_value 0.014. It is true
that r is significantly different from zero, based on the p_value. However, this does not support
the strength. Visit 2 has r = 0.42 with p_value 0.058. With the significance level of 0.05, this
does not support the hypothesis of having non-zero correlation. In other words, this could tell us
that there is no linear relationship between the quality of life and visit 2 measure. The correlation
from the symbol check also has its p_value on the borderline, close to 0.05. I am wondering
about all other domains. Only some correlations were presented or mentioned, and the
correlation between the quality of life and change of cognition scores was not analyzed
appropriately, while the title mentioned this.
I believe that THINC-it could outperform any batteries or tests in many aspects. I’d like to
encourage authors to present the comparison between THINC-it and other methods, regarding
reliability, stability, sensitivity analysis, and power analysis in addition to the correlation
between cognitive functioning change and the quality of life.
Author Response
Dear Reviewer,
Thank you very much for the time taken to review our manuscript titled:
THINC-Integrated Tool (THINC-it): A Brief Measurement Of Changes In Cognitive Functioning And Its Correlation With The Life Quality Of Patients With Schizophrenia And Related Disorders – A Pilot Study.
Please see the attachment where we provide answers to each of your comments. The revisions made by us in the text of the manuscript are highlighted in yellow in the manuscript file as well as in the attached responses.
Dear Reviewers,
Thank you very much for the time taken to review our manuscript titled: THINC-Integrated Tool
(THINC-it): A Brief Measurement Of Changes In Cognitive Functioning And Its Correlation With The
Life Quality Of Patients With Schizophrenia And Related Disorders – A Pilot Study.
Below we provide answers to each of your comments. The revisions made by us in the text of the
manuscript are highlighted in yellow in the enclosed manuscript file as well as in the below
responses.
Reviewer 1:
Review:
Title: THINC-Integrated Tool (THINC-it): A Brief Measurement Of Changes In Cognitive
Functioning And Its Correlation With The Life Quality Of Patients With Schizophrenia And
Related Disorders – A Pilot Study.
Authors: Szmyd, J. K. et al
Table 1 – healthy controls: I did not have a chance to check the description of data, but I am not
sure if two groups are similar to each other for all aspects, except schizophrenia. Otherwise, this
study cannot be an appropriate comparative study. What was the sample size of the control
group?
Response: The control group for the current study serves only for comparison purposes (not
the main analysis, which is based only on the clinical group) and is taken from another study
published by Harrison et al., 2018. This is stated in the manuscript “Data for patients with
schizophrenia participating in the current study (n=21), as well as for healthy controls from a
separate study by Harrison et al. 2018”. Thus, we cannot reliably assess differences between
the clinical and the control group. We have clarified this issue in the Method, Results, and
Limitations subsections (see below).
The number of participants in the control group (Harrison et al., 2018) is 100 participants.
However, our main analysis, as well as main conclusions, are based on within-subject
analysis only based on participants in the clinical group, and control group comparisons
serve only as an additional context for the current findings.
We have added changes to clarify this issue in the Method and Limitations sections, as below:
“As compared with data published for healthy controls from reference study (Table 1) N-back
type task (Symbol Check), the symbol substitution test (Codebreaker), and the Spotter task
posed the most challenge for the patients [28]. ”
“Due to the limitations of the study, we were not able to gather more participants in the
project that would stand as a healthy control group. Therefore to add additional context we
have used data for healthy controls from the reference study of Harrison et al. 2018. We are
aware that this is not fully appropriate to make conclusions, and that ideally, the future
projects should consist of healthy control for proper comparison purposes.”
Another point is, the sample size is too small. With this sample size, authors might lose the
power. It would be better, if the sample size is increased, or if the reason for having a small
sample is justified. I also have a concern on verification of assumptions for hypothesis testing for
the correlations.
Response: The sample size is indeed limited, and should be increased in future studies for
full-scale evaluation of the therapeutic method. The current study is an initial pilot study (as
indicated in the title as well as in the manuscript), and we collected as much participants as
we could in the timeframe allotted for data gathering. The core of the study is based on a
clinical sample that is difficult to collect in large numbers. We have no possibility of
increasing the sample size at this time. Nonetheless, we believe that although the sample size
in our study is limited, it still provides value as an initial pilot study of the THINC-IT
effectiveness. However, we want to be fully transparent about our study limitations, and due
to your suggestion (which we agree with), the limited sample size is now even more strongly
underlined in the Participants and Limitations subsections:
“All subjects were carefully screened and examined for current mental conditions and
severity of symptoms to exclude patients with acute psychotic symptoms, side effects of
pharmacological treatment, or somatic disorders that could negatively influence the cognitive
testing. All patients were introduced to the study procedure by a psychologist and wished to
participate in the study to evaluate their cognitive functions and quality of life. Due to the
exclusion criteria, number of participants was restricted, which, alongside with
heterogeneous nature of the group, added to the limitations of the project.“
“Our ability to analyze obtained data is limited, mainly due to the small number of subjects in
the investigated group and the short observation period. This significantly reduced our ability
to set conclusions about the relationships between cognitive functioning and quality of life. ”
Correlations from the physical health domain: visit 1 had r = 0.53 with p_value 0.014. It is true
that r is significantly different from zero, based on the p_value. However, this does not support
the strength. Visit 2 has r = 0.42 with p_value 0.058. With the significance level of 0.05, this
does not support the hypothesis of having non-zero correlation. In other words, this could tell us
that there is no linear relationship between the quality of life and visit 2 measure. The correlation
from the symbol check also has its p_value on the borderline, close to 0.05. I am wondering
about all other domains. Only some correlations were presented or mentioned, and the
correlation between the quality of life and change of cognition scores was not analyzed
appropriately, while the title mentioned this.
Response: Thank you, we have also added a full correlation table to the manuscript (Table 3),
so the correlation pattern can be evaluated in full.
Results section:
As presented in the Table 3. the number of correct answers in the Codebreaker task at visit 1
is positively related to the quality of life in the Physical health domain at visit 1 (r = 0.053, p =
0.014) and visit 2 (r = 0.42; p = 0.058). The number of correct answers in the Symbol Check task
in the second visit was positively associated with the quality of life in the Environmental
domain, in visit 2 (r = 0.45; p = 0.042). For other measures, better cognitive functioning was
regularly associated with higher quality of life, but the comparisons did not exceed the limit
of statistical significance due to the small number of respondents (n = 21)
Table 3. Correlations for THINC-it cognitive measures and for quality of life scores (WHOQOL-BREF domains) from visit 1 and visit 2 (Pearson's r, p-values in brackets).
|
WHOQOL Domain |
|||||||
Physical health
(visit 1) |
Psychological
(visit 1) |
Social Relationships
(visit 1) |
Environment
(visit 1) |
Physical health
(visit 2) |
Psychological
(visit 2) |
Social Relationships
(visit 2) |
Environment
(visit 2) |
|
Spotter (visit 1)
|
-0.26 |
-0.08 |
0.07 |
0.02 |
-0.35 |
-0.30 |
-0.33 |
-0.01 |
Symbol Check (visit 1) |
-0.03 |
-0.07 |
-0.10 |
0.29 |
-0.00 |
0.15 |
0.26 |
0.45* |
Codebreaker (visit 1)
|
0.53* |
0.15 |
-0.22 |
0.07 |
0.42 |
0.19 |
-0.06 |
0.20 |
Trials (visit 1) |
-0.23 |
0.03 |
0.10 |
-0.01 |
-0.21 |
-0.10 |
-0.24 |
-0.10 |
Spotter (visit 2) |
-0.17 |
-0.03 |
-0.08 |
0.09 |
-0.13 |
-0.09 |
-0.28 |
0.27 |
Symbol Check (visit 2) |
0.12 |
-0.02 |
-0.07 |
0.43 |
0.13 |
0.21 |
0.43 |
0.44* |
Codebreaker (visit 2) |
0.45* |
0.11 |
-0.04 |
0.17 |
0.44* |
0.22 |
0.09 |
0.22 |
Trials (visit 2) |
-0.41 |
-0.15 |
0.08 |
0.07 |
-0.41 |
-0.25 |
-0.27 |
-0.00 |
Note. Significant relationships are bolded.
We have also corrected the statement you refer to, now it correctly evaluates the relationship
at p value of 0.058.
Results section: “Physical health domain at visit 1 (r = 0.53, p = 0.014). The correlation
obtained at visit 2 was non-significant, slightly over the significance level (r = 0.42, p = 0.058).”
“The analysis of the relation between THINC-it tasks and WHOQOL-BREF domains of
quality of life showed that the number of correct responses in the Codebreaker task was
positively associated with the quality of life in the Physical health domain at visit 1 (r = 0.53, p
= 0.014). The correlation obtained at visit 2 was non-significant, slightly over the significance
level (r = 0.42, p = 0.058). “
I believe that THINC-it could outperform any batteries or tests in many aspects. I’d like to
encourage authors to present the comparison between THINC-it and other methods, regarding
reliability, stability, sensitivity analysis, and power analysis in addition to the correlation
between cognitive functioning change and the quality of life.
Response: Thank you, following your valuable suggestion we have added more information
about the psychometric properties of this tool to the manuscript, including the comparison
between THINC-it and other methods of cognitive assesment.
Introduction section:
Furthermore, a separate analysis and healthy controls were conducted to examine the
psychometric characteristics of THINC‐it. To evaluate temporal reliability, stability, and
convergent validity a group of 100 participants completed the full set of cognitive
assessments. For each THINC‐it test (Spotter, Symbol Check, Codebreaker, Trials) an
appropriate comparison test was administered accordingly (Identification Task, OBK -
One‐Back Memory task, DSST - Digit Symbol Substitution Test, TMT-B - Trail Making Test
Part B). This study showed that the THINC-it tool has high levels of reliability (temporal
reliability - Persons’s “r” correlations varying between 0.75 and 0.81; Intrarater reliability - for
all tests ranged between 0.7 and 0.93) and stability (standard deviation values for accuracy
measures ranging from 5.9 to 11.23, and for latency measures ranging from 0.735 to17.3).
Also, levels of convergent validity were in the acceptable range. The correlation between the
Spotter and Identification task of 0.44; Trails and TMT‐B of 0.74; Codebreaker and DSST of
0.63 indicate that those THINCit tasks are a justifiable proxy measure of the tasks from the
comparison assessment. This trend has not occurred between the Symbol Check task and the
OBK - One‐Back Memory task, which resulted in a considerably lower correlation (r = 0.19)
[28].

Reviewer 2 Report
This study aimed to assess the cognitive functioning of patients with the non-validated Polish version of the THINC-it tool and to analyze its association with self-reported quality of life (QOL). It is well written, but we indicate some aspects to consider before its potential publication in a high-impact journal such as Brain Sciences (IF 3.333):
1- The specific objective of the study is to evaluate the cognitive functioning of a sample of patients with schizophrenia through a not validated tool in Polish. On the one hand, the cognitive deficit associated with schizophrenia and its relationship with quality of life is well known; On the other hand, if you want to study the psychometric characteristics of the Polish version of THINC-it, the study design should be different (a validation study).
2- The sample is very small (n = 21), heterogeneous (15 patients with schizophrenia, 2 with bipolar disorder, 1 with depression...). This methodological problem is very serious and prevents any conclusion from being drawn.
3- The assessment is repeated after 6 weeks of intervention. Naturally, if patients improve clinically, they improve in cognitive performance and quality of life. The scientific contribution of this finding is very limited.
4- Since the follow-up period is short (6 weeks), nothing can be inferred about the course of cognitive deficit in a chronic disease.
5- The THIN-it tool provides numerous cognitive variables, but the manuscript tends to highlight those that have shown statistical significance.
Author Response
Dear Reviewer,
Thank you very much for the time taken to review our manuscript titled:
THINC-Integrated Tool (THINC-it): A Brief Measurement Of Changes In Cognitive Functioning And Its Correlation With The Life Quality Of Patients With Schizophrenia And Related Disorders – A Pilot Study.
Reviewer 2
This study aimed to assess the cognitive functioning of patients with the non-validated Polish
version of the THINC-it tool and to analyze its association with self-reported quality of life (QOL).
It is well written, but we indicate some aspects to consider before its potential publication in a
high-impact journal such as Brain Sciences (IF 3.333):
1- The specific objective of the study is to evaluate the cognitive functioning of a sample of
patients with schizophrenia through a not validated tool in Polish. On the one hand, the cognitive
deficit associated with schizophrenia and its relationship with quality of life is well known; On
the other hand, if you want to study the psychometric characteristics of the Polish version of
THINC-it, the study design should be different (a validation study).
Response: Thank you. We have aimed to conduct an initial pilot study of THINC-it efficiency.
In the current work, we have also used a limited set of measures in our study (as to not put
excessive burthen on participants in the clinical sample; results of all of these measures are
reported in the manuscript) and extended analysis of convergent and divergent validity is an
aim for future studies, based on a larger sample of participants. THINC-IT doesn’t contain
subscales that could undergo internal consistency analysis. Saying this, we also agree with
you that extended validation of THINC-IT is needed, which is now stated in the Discussion
section of our work (see below). Simply speaking, we have conducted a study with a slightly
different aim (the pilot study that could evaluate if THINC-IT can be effective for particular
purposes).
“We are convinced that extended analysis of convergent and divergent validity of the Polish
version of the THINC-it tool is needed, and is an aim for future studies - ideally based on a
larger sample of participants with the ability to extend the period of therapeutic interventions
and postponed assessment of patients cognitive performance. ”
2- The sample is very small (n = 21), heterogeneous (15 patients with schizophrenia, 2 with bipolar
disorder, 1 with depression...). This methodological problem is very serious and prevents any
conclusion from being drawn.
Response: As we have also explained in our comment to Reviewer 1, the sample size is
indeed limited, and should be increased in future studies for full-scale evaluation of the
therapeutic method. The current study is an initial pilot study (as indicated in the title as well
as in the manuscript), and we collected as much participants as we could in the timeframe
allotted for data gathering. The core of the study is based on a clinical sample that is difficult
to collect in large numbers. We have no possibility of increasing the sample size at this time.
Nonetheless, we believe that although the sample size in our study is limited, it still provides
value as an initial pilot study of the THINC-IT efficacy. However, we want to be fully
transparent about our study limitations, and due to your suggestion (which we agree with),
the limited sample size is now even more strongly underlined when describing the
participants and in the Limitations subsection:
“All subjects were carefully screened and examined for current mental conditions and
severity of symptoms to exclude patients with acute psychotic symptoms, side effects of
pharmacological treatment, or somatic disorders that could negatively influence the cognitive
testing. All patients were introduced to the study procedure by a psychologist and wished to
participate in the study to evaluate their cognitive functions and quality of life. Due to the
exclusion criteria, number of participants was restricted, which, alongside with
heterogeneous nature of the group, added to the limitations of the project.“
“Our ability to analyze obtained data is limited, mainly due to the small number of subjects in
the investigated group and the short observation period. This significantly reduced our ability
to set conclusions about the relationships between cognitive functioning and quality of life.”
3- The assessment is repeated after 6 weeks of intervention. Naturally, if patients improve
clinically, they improve in cognitive performance and quality of life. The scientific contribution of
this finding is very limited.
Response: Such assumption of a significant relationship between clinical improvement and
improvement of cognitive functions is unfortunately often not true in patients who are not in
the acute exacerbation of schizophrenia. In the present study, only small effect sizes were
found for improvement in three of four THINC-it tasks. This shows that besides standard
treatment the patient should also receive interventions specifically targeting cognitive deficits.
In the manuscript we describe this aspect:
“Furthermore, our study showed only modest improvement in cognitive functions and
quality of life in patients with schizophrenia spectrum disorders after six weeks of standard
care at point-of-care. None of the persons in our study however received experimental
pro-cognition strategies and/or cognitive remediation.”
In answer to the reviewer’s remark, we have also added a statement to the Limitations
subsection highlighting this issue. We agree that future studies evaluating the effectiveness of
THINC-it should employ postponed assessment, preferably several months after the
intervention. At the same time, we think that our design still provides some information
about the efficacy of the method. Other published studies also use a similar design and
provide significant knowledge, although we agree with you that the design would be optimal
with delayed post-measurement.
“To our best knowledge, our pilot study is also the first to document the use of THINC-it as a
cognitive measure in patients with schizophrenia. We are convinced that extended analysis of
convergent and divergent validity of the Polish version of the THINC-it tool is needed, and is
an aim for future studies - ideally based on a larger sample of participants with the ability to
extend the period of therapeutic interventions and postponed assessment of patients
cognitive performance.”
4- Since the follow-up period is short (6 weeks), nothing can be inferred about the course of
cognitive deficit in a chronic disease.
Response: Thank you for pointing our attention to this. We agree that the follow-up period
should be longer to form firm conclusions about the course of cognitive deficits in a chronic
disorder and thus we have devoted attention to making our conclusions more cautious
(multiple changes throughout the manuscript) and added an additional statement about this
to the Limitations subsection:
“Our ability to analyze obtained data is limited, mainly due to the small number of subjects in
the investigated group and the short observation period. This significantly reduced our ability
to set conclusions about the relationships between cognitive functioning and quality of life.
Short follow-up period also reduced our ability to form firm conclusions about the course of
cognitive deficits in a chronic disorder. ”
5- The THIN-it tool provides numerous cognitive variables, but the manuscript tends to highlight
those that have shown statistical significance.
Response: Thank you, we have devoted more space to present insignificant relationships
between analized variables in the Results section. We decided to add a full correlation table to
the manuscript (Table 3), so the correlation patterns can be evaluated in full.
Results section:
As presented in the Table 3. the number of correct answers in the Codebreaker task at visit 1
is positively related to the quality of life in the Physical health domain at visit 1 (r = 0.053, p =
0.014) and visit 2 (r = 0.42; p = 0.058). The number of correct answers in the Symbol Check task
in the second visit was positively associated with the quality of life in the Environmental
domain, in visit 2 (r = 0.45; p = 0.042). For other measures, better cognitive functioning was
regularly associated with higher quality of life, but the comparisons did not exceed the limit
of statistical significance due to the small number of respondents (n = 21)
Please see the attachment where we provide answers to each of your comments. The revisions made by us in the text of the manuscript are highlighted in yellow in the manuscript file as well as in the attached responses.
Table 3. Correlations for THINC-it cognitive measures and for quality of life scores (WHOQOL-BREF domains) from visit 1 and visit 2 (Pearson's r, p-values in brackets).
|
WHOQOL Domain |
|||||||
Physical health
(visit 1) |
Psychological
(visit 1) |
Social Relationships
(visit 1) |
Environment
(visit 1) |
Physical health
(visit 2) |
Psychological
(visit 2) |
Social Relationships
(visit 2) |
Environment
(visit 2) |
|
Spotter (visit 1)
|
-0.26 |
-0.08 |
0.07 |
0.02 |
-0.35 |
-0.30 |
-0.33 |
-0.01 |
Symbol Check (visit 1) |
-0.03 |
-0.07 |
-0.10 |
0.29 |
-0.00 |
0.15 |
0.26 |
0.45* |
Codebreaker (visit 1)
|
0.53* |
0.15 |
-0.22 |
0.07 |
0.42 |
0.19 |
-0.06 |
0.20 |
Trials (visit 1) |
-0.23 |
0.03 |
0.10 |
-0.01 |
-0.21 |
-0.10 |
-0.24 |
-0.10 |
Spotter (visit 2) |
-0.17 |
-0.03 |
-0.08 |
0.09 |
-0.13 |
-0.09 |
-0.28 |
0.27 |
Symbol Check (visit 2) |
0.12 |
-0.02 |
-0.07 |
0.43 |
0.13 |
0.21 |
0.43 |
0.44* |
Codebreaker (visit 2) |
0.45* |
0.11 |
-0.04 |
0.17 |
0.44* |
0.22 |
0.09 |
0.22 |
Trials (visit 2) |
-0.41 |
-0.15 |
0.08 |
0.07 |
-0.41 |
-0.25 |
-0.27 |
-0.00 |
Note. Significant relationships are bolded.

Round 2
Reviewer 2 Report
I appreciate the efforts of the authors to improve the manuscript, but the underlying methodological concerns remain. It is a pilot study, with preliminary and limited findings, which require further study. Taking into account the high impact factor of the journal, I can not reccommend its publication.